# Energy Gap between Doubly Labeled Water-Based Energy Expenditure and Calculated Energy Intake from Recipes and Plate Waste, and Subsequent Weight Changes in Elderly Residents in Japanese Long-Term Care Facilities: CLEVER Study

**DOI:** 10.3390/nu12092677

**Published:** 2020-09-02

**Authors:** Yuki Nishida, Shigeho Tanaka, Satoshi Nakae, Yosuke Yamada, Hiroyuki Shirato, Hirohiko Hirano, Satoshi Sasaki, Fuminori Katsukawa

**Affiliations:** 1Department of Nutrition and Metabolism, National Institute of Health and Nutrition, National Institutes of Biomedical Innovation, Health and Nutrition, Tokyo 162-8636, Japan; nishiday@nibiohn.go.jp (Y.N.); snakae@bpe.es.osaka-u.ac.jp (S.N.); yamada.yousuke@kuas.ac.jp (Y.Y.); 2Sports Medicine Research Center, Keio University, Yokohama 223-8521, Japan; fuminori@keio.jp; 3Faculty of Nutrition, Kagawa Nutrition University, Saitama 350-0288, Japan; 4Division of Bioengineering, Graduate School of Engineering Science, Osaka University, Osaka 560-8531, Japan; 5Institute for Active Health, Kyoto University of Advanced Science, Kyoto 615-8577, Japan; 6Silverpia-Kaga Long-Term Care Health Facility, Tokyo 173-0003, Japan; h-shirato@silverpia-kaga.jp; 7Dentistry and Oral Surgery, Tokyo Metropolitan Institute of Gerontology, Tokyo 173-0015, Japan; h-hiro@gd5.so-net.ne.jp; 8Department of Social and Preventive Epidemiology, Graduate School of Medicine, The University of Tokyo, Tokyo 113-0033, Japan; stssasak@m.u-tokyo.ac.jp

**Keywords:** doubly labeled water, long-term care facility, observed energy intake, total energy expenditure, weight loss

## Abstract

Unintentional weight loss is a major frailty component; thus, assessing energy imbalance is essential for institutionalized elderly residents. This study examined prediction errors of the observed energy intake (OEI) against the actual energy intake obtained from the doubly labeled water (DLW) method and clarified the relationship between the energy gap obtained by subtracting total energy expenditure (TEE) from OEI and subsequent weight changes in elderly residents in long-term care facilities. Overall, 46 participants were recruited in Japan. TEE was measured using the DLW method, and OEI was calculated from recipes and plate waste simultaneously over a 14–15-day period at baseline. The total energy intake (TEI_DLW_) was determined on the basis of DLW and weight changes during the DLW period. The weight was longitudinally tracked monthly for 12 months in the 28 residents who still lived at the facilities. OEI was higher than TEI_DLW_ by a mean of 232 kcal/day (15.3%) among 46 residents at baseline. The longitudinal data of 28 residents showed that the energy gap tended to be correlated with the slope of weight change (*ρ* = 0.337, *p* = 0.080) and the median value was significantly lower in the weight loss group (152 kcal/day) than in the weight gain group (350 kcal/day) (*p* < 0.05). In conclusion, weight loss could occur at Japanese long-term care facilities even if the difference obtained by subtracting TEE from OEI was positive because OEI was overestimated by more than 200 kcal/day.

## 1. Introduction

Weight loss is related to the malnutrition and mortality of elderly residents living in long-term care facilities [1,2]. Weight change depends on the relationship between the total energy intake (TEI) and total energy expenditure (TEE) [3], and body weight decreases if the TEI falls below TEE. Therefore, the appropriate energy requirements of elderly residents need to be set to prevent weight loss. However, the methods used to determine energy requirements in long-term care facilities have not been verified. As a result, the Harris–Benedict equation is still used to predict the basal metabolic rate (BMR) for determining the energy requirement in most Japanese facilities, despite the fact that other BMR equations are more appropriate for older adults [4]. Consequently, the staff at the facilities do not know the appropriate amount of food for the older residents. Therefore, it is important to establish a method for calculating the energy requirements of residents of long-term care facilities.

A previous review showed that weight loss is associated with greater amounts of leftover food, poor oral intake, and feeding dependence in nursing homes [5]. However, even residents who eat all their food often lose weight in the clinical setting. This may be caused by two reasons, in addition to a lack of an appropriate equation for estimating energy requirement. One reason is that actual food provided may be less than the energy of the recipe because the facility staff serve the meal visually. Another reason is that the effect of energy malabsorption is not considered in the calculation of the energy requirement in the present Dietary Reference Intakes for Japanese (2020), although a previous study has suggested that low ingestion and malabsorption are associated with the development of sarcopenia in the elderly population [6]. As such, even if elderly residents eat all the food provided by facilities, their weight may decrease as a result of a low TEI.

The doubly labeled water (DLW) method is often used as a reference method for examining the validity of TEI by dietary assessments. This method is based on an assumption that the TEE measured using the DLW method should be equal to the TEI if weight change does not occur [7]. In contrast, if weight change is expected during the DLW period, the value adjusted by adding the change in body mass or composition (energy stores) to TEE is closer to the actual TEI [8]. To the best of our knowledge, only Persson et al. have simultaneously measured TEE using the DLW method and TEI by dietary assessment in elderly residents at nursing homes [9]. However, this study had two potential issues with regards to the outcome. First, they showed that the TEI, determined on the basis of dietary records, was overestimated by approximately 8%, but the weight change during the DLW period was not considered in the analysis, and it could not be determined whether the difference between the TEE and the calculated TEI was simply overestimation or actual overfeeding. Second, they did not report the relationship between the energy gap and subsequent weight change; thus, it is unclear whether some portion of the energy gap could be explained by weight changes. It is important to identify the magnitude of the errors that may occur when calculating the observed energy intake (OEI) of residents using the energy content of recipes and plate waste, and set energy requirement after considering such errors to prevent weight loss for elderly residents living in long-term care facilities.

This paper reports some of the results of the Clinical Evaluation of Energy Requirements (CLEVER) study. The aims of this paper were to (1) calculate the error of estimating the OEI from recipes and plate waste using the DLW method, and (2) clarify the relevance between the energy gap (OEI minus TEE) and the subsequent change in weight in older residents at several long-term care facilities.

## 2. Materials and Methods

### 2.1. Protocol

The TEE was evaluated using the DLW method in each long-term care facility over a 14–15-day period. On the first and last day, weight was measured in the fasted state, and an oral dose of DLW was given on the first day. During the baseline period, the BMR was measured before breakfast, and all dietary records and recipes were collected. Blood samples were also obtained to examine nutritional status, and the subjects and facility staff were asked to answer questionnaires about the subjects. The follow-up weight measurements were performed at 6 months, and then 12 months later, while monthly weight data were obtained from clinical records up to 1 year after the baseline survey.

The study protocol was approved by the Ethics Committees of Keio University (no. 2015-03) and the National Institutes of Biomedical Innovation, Health, and Nutrition (No. NIBIOHN29) in accordance with the Declaration of Helsinki. All participants and their relatives gave written informed consent after being advised of the study procedures.

### 2.2. Subjects

In total, 64 older residents who could eat without assistance were recruited from five long-term care facilities in metropolitan Tokyo, Japan. The exclusion criteria were outlined in the previous article [10]. In this study, some participants were excluded from the analyses in accordance with the two aims of the study; a flowchart of the exclusion criteria is shown in Figure 1. We have previously published a paper including 58 older residents in the baseline survey [10]; we then excluded 11 participants who left >10% of their food to minimize the errors of plate waste by visual estimation, and one participant who used ≥3 suppositories during the baseline survey (analysis 1). In addition, 18 participants were excluded due to marked plate waste (*n* = 1), death (*n* = 1), hospitalization (*n* = 8), or moving facilities (*n* = 8) during the follow-up period (analysis 2).

### 2.3. Anthropometry

Researchers measured the weight of subjects with light clothes, no shoes, and empty pockets to the nearest 0.1 kg using an electronic scale at each facility. The DLW method is quite expensive to implement and thus we could not conduct a follow-up survey by the same measurement as baseline. Thus, body weights had been followed up to examine whether energy served was appropriate for elderly residents. Weight was measured on the first and last day of the baseline survey and 6 months and 12 months later. Monthly weight data were also obtained from clinical records that had been filled out by the staff at each facility. Because nearly 50% of the subjects could not remain in a standing position, the stature was obtained from medical records, but there was no information on how the height was measured or estimated, i.e., interview or measuring tape. Body mass index (BMI) was calculated as body weight divided by height squared (kg/m^2^).

### 2.4. Calculation of Observed Energy Intake (OEI)

All recipes and records of visual plate waste by facility staff during the DLW period were collected from each facility. The estimated energy contents of recipes were determined from the Standard Tables of Food Composition in Japan 2015 (Seventh Revised Edition) [11]. The energy requirement at admission into the facility was determined as follows: predicted BMR, calculated by the Harris–Benedict equation and physical activity level (PAL), or activity and stress factors, body weight and a coefficient of 20–30 (kcal/kg/day). In addition, three facilities set lower energy requirements for residents with diabetes or obesity. The proportion of leftover food was estimated using the quintile method or decile method. OEI was defined as the energy served by the recipe minus the estimated visual plate waste and was calculated by a skilled dietitian in our research group. Nutritional composition analysis of the diets was conducted using a nutrient computer software program (Eiyo-kun Ver.6.0, KENPAKUSHA, Tokyo, Japan). Detailed information on the dietary survey is shown in the Appendix A.

### 2.5. Measurement of Energy Expenditure

The TEE was measured using the modified two-point DLW method, the details of which have been described elsewhere [10]. Briefly, the dose of DLW was given on the morning after a baseline (BL) blood sample had been collected. An oral dose of 0.1 g ^2^H_2_O and 2.0 g H_2_^18^O per kilogram was administered according to the estimated total body water. Post-dose blood samples were collected approximately 3 h (PD3) and 4 h (PD4) after administration at the same time on days 14 or 15 (ED1 and ED2). The plasma was separated by centrifuging for 15 min at 4 °C and then transferred to dry and then tightly sealed in containers for immediate freezing to −30 °C. Isotope analysis of the blood samples was performed in duplicate using an isotope ratio mass spectrometer (Hydra 20–20 Stable Isotope Mass Spectrometers; Sercon Ltd., Crewe, UK). The ^2^H/^1^H ratios were analyzed by hydrogen gas equilibration using a platinum catalyst, and the ^18^O/^16^O ratios were determined after carbon dioxide equilibration. The isotope analyses were conducted at ESTech Kyoto Corporation, Ltd. (Shiga, Japan). The average standard deviation for these analyses was 1.0‰ ± 0.8‰ for ^2^H, and 0.09‰ ± 0.09‰ for ^18^O. The total body water was calculated as the mean of the dilution space estimated by ^2^H and ^18^O after correction for isotope exchange by 1.041 and 1.007, respectively [12].

Carbon dioxide production was estimated from the difference between the elimination rates of ^2^H and ^18^O, and was used to calculate TEE [13]. The food quotient was calculated for each individual using dietary data and the Black equation [14].

The BMR was measured by indirect calorimetry using a ventilated hood (Quark RMR, COSMED, Rome, Italy). The gas exchange measurement was initiated after the subject had rested comfortably for 15–20 min lying down, and consistent data longer than 5 min were used in the analyses. The BMR was calculated according to the Weir equation [15], and the PAL was calculated as the TEE divided by the BMR.

### 2.6. Calculation of Total Energy Intake Determined Using the DLW Method (TEI_DLW_)

To compare the OEI with the DLW method, we calculated the TEI_DLW_ from the TEE and the change in weight during the baseline survey. The Bathalon equation [16] was used to calculate the TEI_DLW_ as follows:TEI_DLW_ = TEE + (⊿weight × 7)(1)
where ⊿weight was measured as g/day between the first and last day during the baseline survey, and 7 (kcal/g) is the energy density of the change in weight [17].

### 2.7. Questionnaires

Several questionnaires were conducted to determine the characteristics of subjects. Functional ability was evaluated using the Barthel Index (BI) [18]; this questionnaire comprises 10 items (feeding, grooming, bathing, dressing, bowel and bladder care, toilet use, ambulation, transfers, and stair climbing), with a total score ranging from 0 to 100. Higher scores represent greater independence in performing daily living activities. The Mini Nutritional Assessment—Short Form (MNA-SF) was used to evaluate nutritional status [19,20]; this form includes the decline in food intake, weight loss in the previous 3 months, mobility, acute disease/distress, psychological situation, and BMI. The MNA-SF has a three-category scoring system: 14 to 12 indicates normal nutritional status, 11 to 8 indicates a risk for undernutrition, and 7 to 0 indicates undernutrition. The Rapid Geriatric Assessment (RGA), comprising four screening tests, was administered [21]. The FRAIL (Fatigue, Resistance, Aerobic, Illnesses and Loss of weight) scale, with scores ranging from 0 to 5, to identify frailty, with a higher score indicating increased frailty, was also administered. In the FRAIL scale, a score of zero indicates a healthy elderly adult, 1–2 indicates pre-frail or early decline, and 3 or greater indicates decline and frailty. The SARC-F (Strength, Assistance in walking, Rise from a chair, Climb stairs and Falls) scale for sarcopenia has good predictive ability for future dependency in activities of daily living; the scores range from 0 to 10, with a score of 4 or greater indicating sarcopenia. The Simplified Nutritional Appetite Questionnaire (SNAQ) has been validated in various elderly people, and a score of 14 or less is predictive of weight loss over the following 6 months. The rapid cognitive screen (RCS) is the shortened version of the well-validated Saint Louis University Mental Status Examination, with scores ranging from 0 to 10, with 0–5 indicating dementia, 6–7 indicating mild cognitive impairment, and 8–10 representing a normal score. Facility staff answered the BI and MNA-SF, while the subjects answered the RGA. The BI, the FRAIL scale, and the SARC-F scale are related to physical activity and energy expenditure while the MNA-SF and SNAQ are related to the energy intake.

### 2.8. Calculations and Statistical Analyses

The Shapiro–Wilk test was used to evaluate the normality of the data. Data of analysis 1 are expressed as mean ± SD, and data of analysis 2 are expressed as median and range (min–max). Differences between the energy variables in analysis 1 were compared using the paired *t*-test, and the agreement between OEI and TEI_DLW_ was assessed using Bland and Altman plots [22]. The limits of agreement were plotted and defined as 1.96 SD of the difference above and below zero. Before the statistical analysis of analysis 2, it was necessary to clarify two assumptions. First, facility staff measure the weight for residents every month in Japanese long-term care facilities. However, it is unclear how accurately these weights are measured because there is no manual for measurement of weight and their weight is assumed to be often measured with or without something in their pocket, wearing shoes, and/or wearing a heavy jacket. Therefore, researchers also measured the weight of residents with light clothes, no shoes, and empty pockets, and we needed to compare both weights to examine the accuracy of the weight measured by facility staff. The inter-rater reliability of the weight measurement between the researcher and facility staff was examined by intra-class correlation coefficient at baseline and 6 months and 12 month later. Second, the subjects in analysis 2 were followed up from baseline to 12 months and had no acute exacerbations or marked plate waste. It was hypothesized that the weight would change at a constant rate; therefore, the slope of the regression line of weights measured once a month over 12 months was used to assess the weight changes, where a negative and positive slope defined weight loss and gain, respectively (Figure 2). We also calculated the root mean square error of variation (RMSE) to support the above hypothesis. The RMSE is used as an indicator of weight variability [23] and calculated around each resident’s regression line for monthly weight changes in the present study. In the second analysis, the relationship between the energy gap and the slope of weight change was examined by Spearman’s rank correlation coefficient. Differences in variables between the weight loss and gain groups were compared using the Mann–Whitney *U* test. All statistical analyses were performed using SPSS for Windows 15.0 (SPSS Inc., Chicago, IL, USA). Significance was set at *p* = 0.05.

## 3. Results

### 3.1. Analysis 1: Prediction Error of OEI

A total of 46 participants were included in analysis 1, where errors in estimating the OEI using the DLW method were examined. The BI score was 68.3 ± 19.8, and the MNA-SF was 9.8 ± 1.9; 36 subjects (78.3%) were classified as at risk of undernutrition, and 3 subjects (6.5%) were classified as undernourished. The albumin levels were 3.8 ± 0.4 (g/dL), and 37.0% of subjects had levels below standard (3.8 (g/dL). The pre-albumin levels were 19.1 ± 4.7 (g/dL), and 73.9% of the subjects had levels below 22.0 (g/dL).

Table 1 shows the physical characteristics and energy variables of the subjects in analysis 1. The average value (min–max) of the plate waste rate was 1.6% (0.0–7.4), which indicated that most subjects ate all of the meals provided by the facility. The TEE was 1159 ± 190 kcal/day (26.0 ± 3.5 kcal/kg/day), and the TEI_DLW_ was 1282 ± 281 kcal/day (29.1 ± 6.9 kcal/kg/day) by considering weight changes during the DLW period. The OEI was overestimated compared to the TEI_DLW_ by a mean of 232 kcal/day (15.3%), which was significantly different to zero (Figure 3).

### 3.2. Analysis 2: Relationship between the Energy Gap at Baseline and Subsequent Body Change

The data of 28 subjects were analyzed to examine the relationship between the energy gap (OEI-TEE) at baseline and the subsequent change in weight during the 1 year follow-up period. All body weights measured by the facility staff at baseline and 6 months and 12 months later were a little higher than those measured by the researchers; the differences in body weight were 0.54 ± 1.52 (kg), 0.50 ± 0.65 (kg), and 0.80 ± 1.31 (kg), respectively. The intra-class correlation coefficients indicated good reliability (ICC = 0.977, 0.996, and 0.980, respectively); thus, body weight recorded by the facility staff is considered to be reliable. The RMSE values of the subjects ranged from 0.31 to 1.52. The case who was excluded, as outlined above, had a RMSE of 2.80 (Figure 2c). Therefore, in comparison, the subjects in analysis 2 were considered to have no rapid change in weight. The energy gap tended to be correlated with the slope of weight change (*ρ* = 0.337, *p* = 0.080) (Figure 4). Although the difference obtained by subtracting TEE from OEI was positive in most subjects (*n* = 27), seven older residents lost weight after 1 year, despite the fact that their energy gaps were positive.

Table 2 shows the comparisons of baseline characteristics between the weight loss and weight gain groups. Of the 28 subjects who were followed up for 1 year, 8 (3 men and 5 women) were classified as the weight loss group and 20 (3 men and 17 women) were classified as the weight gain group. The height and body weight at baseline were significantly higher in the weight loss group than in the weight gain group (*p* < 0.05). However, there was no significant difference in BMI between the two groups (*p* = 0.476). The amount of energy served was changed for two people in each group. Among the weight loss group, one resident increased energy served by 200 kcal/day because he walked around all day, and the other resident decreased energy by 250 kcal/day due to her wish. In contrast, in the weight gain group, one resident increased energy by 150 kcal/day with the change of diet type, and another decreased energy by 200 kcal/day for weight control. There were no significant differences for admission period to baseline (*p* = 0.401), number of medicines (*p* = 0.405), and complication (*p* = 0.138) between the weight loss and weight gain groups. The SNAQ scores tended to be lower in the weight gain group than in the weight loss group (*p* = 0.051), while seven subjects were classified as having a poor appetite in the weight gain group compared to only one subject in the weight loss group. The other questionnaires and blood markers did not show the significant difference between both groups.

Table 3 shows the energy served, OEI, TEE, BMR, and PAL. The energy served (kcal/kg/day) was significantly lower in the weight loss group than in the weight gain group (*p* < 0.01). The median value of the difference obtained by subtracting TEE from OEI was significantly lower in the weight loss group than in the weight gain group by approximately 200 kcal/day (*p* < 0.05). Meanwhile, there was no significant difference for OEI-TEI_DLW_ between weight loss and weight gain groups (*p* = 0.204). BMR was significantly higher in the weight loss group than in the weight gain group (*p* < 0.05), while there was not significant difference for PAL (*p* = 0.899). The baseline characteristics of each facility are shown in the Appendix A.

## 4. Discussion

The current study shows that errors may occur in the measurements of OEI, and in the relationship between the energy gap and changes in weight in elderly residents at Japanese long-term care facilities. The results showed that the OEI was higher than the TEI_DLW_ by more than 200 kcal/day, and that weight loss occurred even if the energy gap calculated by subtracting TEE from OEI was positive. As an additional finding, although both groups had a positive energy gap on average, the energy gap was significantly lower in the weight loss group compared to the weight gain group, and tended to be correlated with the slope of weight change. Moreover, height and weight, which related to BMR, were significantly higher in the weight loss group than in weight gain group; thus, the estimated energy requirement may be insufficient for residents who had larger body sizes.

Previous studies have shown that the average rate of plate waste was greater than approximately 10% in elderly residents in long-term care facilities [24,25]. Another study reported that severe weight loss occurred when elderly residents ate less than 75% of their meals [26]. In contrast, the present subjects could eat almost all of their meals independently and survived for 1 year without acute disease and marked plate waste. Considering the above, our subjects seemed to be in a more stable condition than the subjects of previous studies. However, 70–80% of subjects were classified as being at risk for under nutrition or malnutrition from the results of the questionnaires and blood analyses, and a significant number of subjects lost weight after 1 year. These results suggest that the OEI calculated by current methods may not be equivalent to the actual TEI.

Engelheart and Akner investigated the TEI from weighed food records and showed that the average ± SD TEI was 27 ± 8 kcal/kg/day [27], while Buckinx et al. reported the mean TEI as 25.7 kcal/kg/day [25]. Our results showed that OEI and TEI_DLW_ were 34.5 ± 6.7 kcal/kg/day and 29.1 ± 6.9 kcal/kg/day, respectively, which are higher than those reported previously. This difference can be attributed to the fact that our subjects were relatively active residents at each facility, and were able to eat most meals, whereas the subjects in previous studies required nursing care to carry out several daily activities, and frequently had leftovers. Additionally, the lower TEI in the current study was due to the low weight of our subjects, since TEI per kilogram was negatively correlated with weight [28].

Our finding suggests that the OEI calculated in long-term care facilities may overestimate their actual TEI by more than 200 kcal/day as compared to the TEI_DLW_ in our cohort. As a result, the median energy gap was positive by 152 kcal/day, even in the weight loss group; the estimation error of plate waste could be one of the reasons for this overestimation. Several weighing methods and visual estimation methods are used to calculate plate waste [29]; in this study, plate waste was calculated by visual estimation, which is often used in the clinical situation. Although the degree of error of visual estimation methods compared to weighing methods remains unclear, there is some evidence that greater plate waste causes greater errors in visual estimation [30,31]. In most cases, the subjects in our study ate without leaving plate waste; therefore, the errors in visual estimation would have been minimal. Accordingly, a possible reason for the observed discrepancy between OEI and TEI_DLW_ is that not all of the food described in the recipe was served to the residents. Consequently, errors in energy served occur in the current system because the main and side dishes are usually served without being weighed [32]. Thus, future studies are necessary to confirm the accuracy of the methods used to determine the amount of meals served.

Another possible reason for the large discrepancy between OEI and TEI_DLW_ is malabsorption in the older population. Although studies have suggested that malabsorption leads to sarcopenia or frailty, it is not known to what degree the malabsorption rate increases with aging. In fact, malabsorption is not considered by any facility when calculating the energy requirements of residents. Many animal studies have examined excretory energy loss. Raman et al. showed a significant 4% decrease in absorption efficiency in rhesus monkeys under long-term calorie restriction [33]. The author raised the possibility that endogenous sources have the potential to contribute to increases in energy loss, although this possibility or other potential reasons have yet to be validated. In contrast, Wierdsma et al. showed that intestinal energy absorption was approximately 90% in healthy human adults [34]. Other studies have also suggested that low digestibility of energy is caused by a history of either malabsorption [35], short bowel syndrome [36], or exocrine pancreatic insufficiency [37]. Malabsorption may have occurred in our subjects because they had symptoms of chronic constipation and diarrhea.

The base characteristics of the two groups were almost the same. Therefore, weight loss was more affected by the lack of energy served than by participants’ characteristics. However, only SNAQ scores tended to be lower in the weight gain group than in the weight loss group. SNAQ was originally developed from the Council of Nutrition Appetite Questionnaire (CNAQ), which was validated in nursing home residents [38]. SNAQ, an index of appetite, not only predicts weight loss over a 6-month period, but is also inversely associated with 1-year mortality [39]. The current study showed that the SNAQ scores were higher in the weight loss group than in the weight gain group. All subjects in the current study could eat independently without plate waste and were a relatively more active population than subjects in the earlier studies described above. The amount of food in a meal was fixed in long-term care facilities, and therefore, the possibility that the TEE exceeds the actual TEI needs to be considered. For example, if a resident does not feel full even after eating all their meal, it may be necessary to reconsider the amount of food that is provided to them.

The present study focused on people who could eat almost all of their meals at baseline and survived for 1 year without acute disease and marked plate waste. The main strength of our study is the ability to examine the relationship between the energy gap and weight changes. However, there are some limitations. First, the sample size was small, with 90% of residents excluded at baseline because they were using medications that affected energy or water metabolism. In addition, Japanese long-term care facilities define an admission period as 3 months in principle, despite not being used in practice. Therefore, a follow-up survey over a long period was difficult for residents in long-term care facilities. Second, we did not calculate the measurements using a precise weighing method, which might have resulted in a large difference between the OEI and the TEE. However, because all the subjects ate nearly all of their meals during the study, the error in visual estimation of plate waste would have been minimal. Third, the subjects in our study could perform daily activities in the facility relatively independently; future studies should focus on subjects who require more nursing care. Fourth, our group did not measure energy loss into stool and urine by bomb calorimeter. Residents seldom leave the facility and often use portable toilet, and their meals can be obtained from facility; thus, we consider it feasible to collect their stool and urine, and measure energy loss among elderly residents at these facilities in the future study. Fifth, we could not follow the energy gap because the DLW is quite expensive. Meanwhile, we validated the simple physical activity record for estimating TEE compared with the DLW method in the previous paper [10]. If use that simple record instead of the DLW method, we can increase sample size and examine the true energy requirement that takes into account the underestimation of OEI.

## 5. Conclusions

In conclusion, the OEI calculated from plate waste and recipes used at Japanese long-term care facilities was overestimated by more than 200 kcal/day compared to the TEI_DLW_. Therefore, weight loss could occur even if residents seemed to eat more than their TEE. In particular, the more energy served needs to be provided for residents who had larger body size. The present results are based on limited data from five facilities, and further studies at multiple facilities are needed to discuss whether the current calculation method of OEI is appropriate for preventing weight loss.

## Figures and Tables

**Figure 1 nutrients-12-02677-f001:**
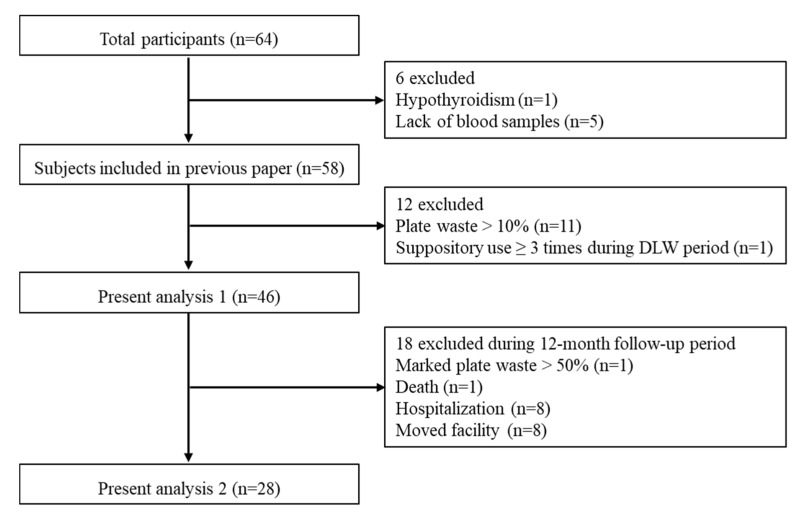
Flowchart of the exclusion criteria for the study.

**Figure 2 nutrients-12-02677-f002:**
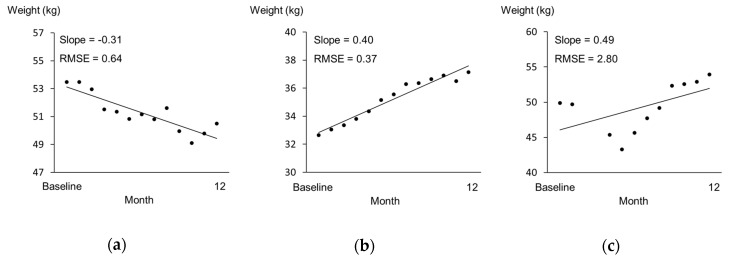
Samples for monthly weight change. The *x*-axis represents months since baseline, and the *y*-axis represents weight (kg). (**a**) A subject in the weight loss group. (**b**) A subject in the weight gain group. (**c**) A subject excluded due to their re-entry to the facility after hospitalization in the middle of the follow-up period.

**Figure 3 nutrients-12-02677-f003:**
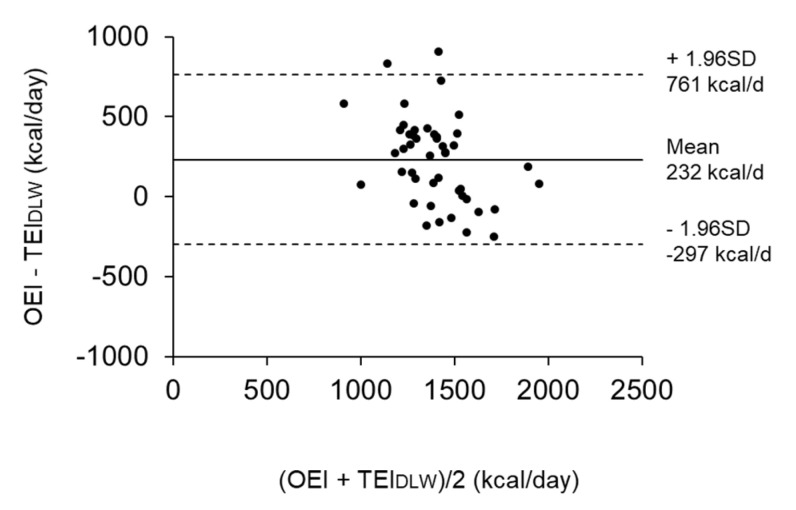
Bland Altman plots showing the difference between the observed energy intake (OEI) and total energy intake determined using the doubly labeled water (DLW) method (TEI_DLW_) in reference to the mean (*n* = 46). The solid line shows the mean difference, while the dotted lines represent the limits of agreement (± 1.96 SD) from the mean difference.

**Figure 4 nutrients-12-02677-f004:**
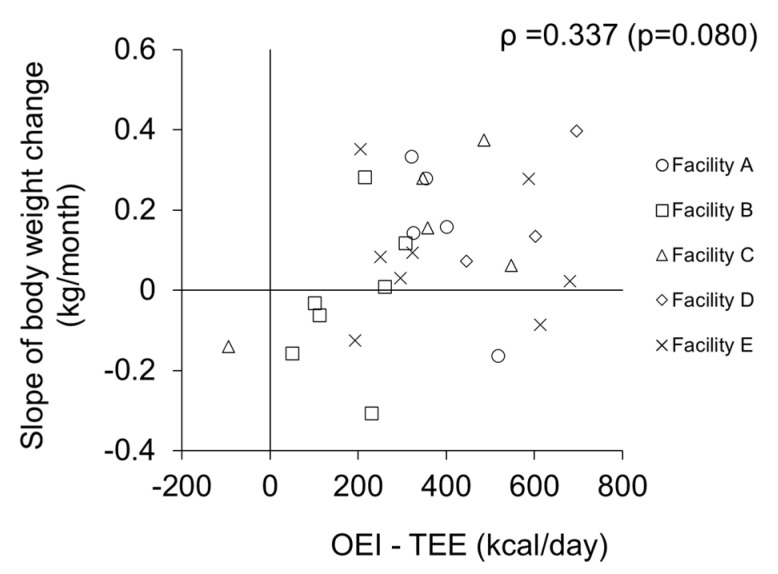
Relationship between energy gap and slope of weight change (*n* = 28).

**Table 1 nutrients-12-02677-t001:** Baseline characteristics and energy variables (analysis 1).

		Total (*n* = 46)		
Age	(year)	85	±	7
Female	*n* (%)	32 (69.6)		
Height	(cm)	150.2	±	9.6
Weight	(kg)	44.9	±	7.2
⊿Weight	(kg)	0.2	±	0.4
BMI	(kg/m^2^)	19.6	±	3.7
TEE	(kcal/day)	1159	±	190
	(kcal/kg/day)	26.0	±	3.5
TEI_DLW_	(kcal/day)	1282	±	281
	(kcal/kg/day)	29.1	±	6.9
Energy served	(kcal/day)	1536	±	182
	(kcal/kg/day)	35.0	±	6.7
Rate of plate waste	(%)	1.5	±	2.1
OEI	(kcal/day)	1514	±	187
	(kcal/kg/day)	34.5	±	6.7
OEI-TEI_DLW_	(kcal/day)	232	±	270
BMR	(kcal/day)	863	±	126
PAL		1.35	±	0.14

Values are expressed as mean ± SD. ⊿Weight: body weight change during the doubly labeled water period; BMI: body mass index; TEE: total energy expenditure measured by the doubly labeled water method; TEI_DLW_: TEI determined using the doubly labeled water method; OEI: observed energy intake calculated from energy of served menu and percentage of plate waste; BMR: measured basal metabolic rate; PAL: physical activity level calculated as the measured total energy expenditure divided by the measured basal metabolic rate.

**Table 2 nutrients-12-02677-t002:** Baseline characteristics and comparison between the weight loss and gain groups (*n* = 28).

		Weight Loss (*n* = 8)		Weight Gain (*n* = 20)		*p*-Value
Age	(year)	86	(71–99)	87	(75–94)	0.524
Female	*n* (%)	5	(63)	17	(85)	
Height	(cm)	153.1	(148.0–172.0)	145.0	(129.0–167.5)	0.008
Weight	(kg)	47.1	(42.4–61.5)	43.3	(31.3–54.8)	0.024
BMI	(kg/m^2^)	20.2	(17.8–23.4)	19.7	(16.1–23.3)	0.476
Change in energy served during follow-up period	*n* (%)	2	(25)	2	(10)	
Admission period to baseline	(months)	9.1	(6.3–63.9)	16.1	(4.3–69.5)	0.401
Number of medicines		4	(1–8)	3	(0–7)	0.405
Number of complications		4	(2–6)	5	(2–10)	0.138
BI	(score)	75	(50–95)	78	(30–100)	0.838
MNA-SF	(score)	10	(6–11)	10	(6–12)	0.424
Risk for under nutrition	*n* (%)	7	(88)	16	(80)	
Malnutrition	*n* (%)	1	(13)	1	(5)	
FRAIL scale, score		2	(0–3)	2	(0–4)	0.792
Pre-frail, no. (%)		4	(50)	13	(65)	
Frail, no. (%)		2	(25)	4	(20)	
SARC-F, score		4	(0–6)	4	(0–9)	0.522
Risk for sarcopenia, no. (%)		4	(50)	11	(55)	
SNAQ, score		16	(13–18)	15	(11–17)	0.051
Poor appetite, no. (%)		1	(12.5)	7	(35)	
RCS score		4	(1–7)	5	(0–10)	0.571
Mild cognitive, no. (%)		2	(33)	3	(19)	
Dementia, no. (%)		6	(75)	15	(75)	
Serum albumin	(g/dL)	3.8	(3.4–4.3)	3.9	(3.3–4.4)	0.645
Serum pre-albumin	(g/dL)	19.6	(12.7–27.7)	21.4	(13.7–35.4)	0.576

All values are expressed as median (min to max). BMI: body mass index; BI: Barthel Index; MNA-SF: Mini Nutritional Assessment-Short Form; SARC-F: Strength, Assistance in walking, Rise from a chair, Climb stairs and Falls; SNAQ: Simplified Nutritional Appetite Questionnaire; RCS: rapid cognitive screen. *p*-values were calculated by the Mann–Whitney *U* test for comparison between the weight loss and gain groups.

**Table 3 nutrients-12-02677-t003:** Energy served, observed energy intake, energy expenditure, and physical activity level (analysis 2).

		Weight Loss (*n* = 8)		Weight Gain (*n* = 20)		*p*-Value
Energy served	(kcal/day)	1426	(1268–1779)	1558	(1050–1984)	0.146
	(kcal/kg/day)	29.1	(25.3–42.0)	36.4	(26.4–50.9)	0.006
OEI	(kcal/day)	1385	(1263–1779)	1511	(1037–1984)	0.154
	(kcal/kg/day)	28.2	(25.2–42.0)	35.4	(26.3–50.9)	0.005
TEE	(kcal/day)	1189	(904–1642)	1110	(777–1416)	0.093
	(kcal/kg/day)	25.2	(18.6–27.6)	25.6	(21.1–32.8)	0.557
OEI-TEE	(kcal/day)	152	(−94–614)	350	(205–696)	0.022
	(kcal/kg/day)	3.3	(−1.5–14.2)	8.9	(4.2–21.3)	0.012
OEI-TEI_DLW_	(kcal/day)	59	(−179–514)	294	(−159–835)	0.204
	(kcal/kg/day)	1.3	(−3.6–12.1)	7.3	(−3.6–18.1)	0.119
BMR	(kcal/day)	905	(787–1046)	783	(631–1119)	0.050
	(kcal/kg/day)	18.4	(17.0–19.5)	19.7	(15.9–21.9)	0.082
PAL		1.34	(1.04–1.57)	1.36	(1.10–1.58)	0.899

All values are expressed as median (min to max). OEI: observed energy intake; TEE: total energy expenditure by the doubly labeled water method; TEI_DLW_: total energy intake estimated by the doubly labeled water; BMR: measured basal metabolic rate; PAL: physical activity level calculated as the measured total energy expenditure divided by the measured basal metabolic rate. *p*-values were calculated using the Mann–Whitney *U* test for comparison between the weight loss and gain groups.

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
