# Peer review of "Energy Gap between Doubly Labeled Water-Based Energy Expenditure and Calculated Energy Intake from Recipes and Plate Waste, and Subsequent Weight Changes in Elderly Residents in Japanese Long-Term Care Facilities: CLEVER Study"

_nutrients, 2020, doi:10.3390/nu12092677_

Round 1
Reviewer 1 Report
Summary: The authors studied the gap between observed energy intake and energy intake and total energy expenditure measured by doubly labeled water in elderly, nursing home residents. They report that observed energy intake was 232 kcal/d greater than DLW-measured energy intake in 46 subjects at baseline. They also report that the energy gap between observed intake and expenditure tended to be correlated with weight change over 1 year in a subset of 28 subjects, and that the overestimation in observed intake compared to measured expenditure was lower for those who lost weight than those who gained weight. The research was performed by a group with an excellent track record in the field of energy expenditure measurement and this is reflected in their methodological approaches. The manuscript is generally well-written, but I have several suggests below that may improve the readability of the manuscript.
Comments:
L30: The authors should clarify the time period for “baseline” measurements in the abstract.
L33: It is unclear which results in the abstract refer to measurements of the initial 46 participants versus the 28 followed longitudinally.
L34: “and the energy gap was significantly lower in the weight loss group than in the weight gain group (p < 0.05).” This may be clearer if the numeric values of the energy gap are reported, since it was positive for both groups but that was not stated.
L36: The abstract lacks a conclusion.
L53: “This may explain errors in the calculation of energy intake and malabsorption specific to older people, in addition to a lack of energy requirement.” I had difficulty interpreting this sentence. Do the authors mean that greater malabsorption in older people is not taken into account by the equations used to predict energy intake and, thus, the food provided is below their energy requirements?
L69: “First, they showed that the EI, determined based on dietary records, was overestimated by approximately 8%, but the weight change during the DLW period was not considered in the analysis, and it was unclear a discrepancy existed in the TEI. Second, they did not report the relationship between the energy gap and subsequent weight change; thus, it could not be determined whether the difference between the TEE and the calculated TEI was simply overestimation or actual overfeeding.” I am having trouble understanding the difference between the two points that the authors are trying to make here. Do the authors mean here that EI by food records was 8% higher than the TEE by DLW, but, since weight change was not reported, it is not clear whether some portion of this difference could be explained by weight gain (in the case of overfeeding) or whether the true difference between observed EI and measured EE may have actually been greater due to weight loss?
L186: “First, the inter-rater reliability of the weight measurement between the researcher and facility staff was examined by intra-class correlation coefficient at baseline and 6 months and 12 month later.” Can the authors be more specific about how this was done? I assume the authors had both staff and researcher weigh all of the subjects included in analysis 2 at these time points, but it would be good to state that.
L190: “It was hypothesized that the weight would change at a constant rate; therefore, the slope of the regression line of weight change over 12 months was used to assess the weight changes…” Can the authors be more specific about how often subjects were weighed? From Figure 2 it appears that they were weighed monthly, but some statement of this would help clarify.
L192: “We also calculated the root mean square error of variation (RMSE) to support the above hypothesis.” By RMSE, are the authors referring to the RMSE between the regression line and the measured values? (as opposed to the baseline weight or mean weight or the 12-month period).
L194: “In the second analysis, the relationship between the energy gap and the slope of weight change was examined by Pearson's correlation coefficient.” Since all other analyses in the second group were non-parametric (presumably due to lower numbers) would it be more appropriate to use Spearman Rho instead of Pearson correlation (although the difference is likely small)?
L214: It should be noted that the plate waste rate data is in Table 1.
Table 2: the units for BMR should be kcal/d rather than kcal/kg/d.
Table 3: Can the authors elaborate on “Change in energy served during follow-up period”? Does this mean that the amount of energy served was changed for 2 people in each group? And in what way was it changed?
L252: It may help the interpretation of the results to report the difference in OEI vs DLW-EI for the subset that were reported on in analysis 2 and by group in analysis 2 – i.e. was the overestimation of EI by observation similar at baseline for weight loss and weight gain groups?
L269: “…and that the relationship between the energy gap and weight changes was not significant” The authors send a somewhat mixed message here, since in the abstract they note that “The energy gap tended to be correlated with the slope of weight change” and they point out that the energy gap is lower for weight loss group both in abstract and discussion.
L340: The authors could add the lack of measurement to determine malabsorption (stool and urine collection) and lack of follow-up measurements for TEE and OEI to limitations.
Author Response
Response to Reviewer 1 Comments
Reviewer 1
Summary: The authors studied the gap between observed energy intake and energy intake and total energy expenditure measured by doubly labeled water in elderly, nursing home residents. They report that observed energy intake was 232 kcal/d greater than DLW-measured energy intake in 46 subjects at baseline. They also report that the energy gap between observed intake and expenditure tended to be correlated with weight change over 1 year in a subset of 28 subjects, and that the overestimation in observed intake compared to measured expenditure was lower for those who lost weight than those who gained weight. The research was performed by a group with an excellent track record in the field of energy expenditure measurement and this is reflected in their methodological approaches. The manuscript is generally well-written, but I have several suggests below that may improve the readability of the manuscript.
We would like to thank Reviewer 1 for the insightful comments and constructive suggestions. We have revised the manuscript per the reviewer’s comments.
Our responses to the reviewer’s comments are as follows:
Q1. L30: The authors should clarify the time period for “baseline” measurements in the abstract.
A1. We have added the time period for “baseline” measurements (Page 1, Line 30-31) : New text
Q2. L33: It is unclear which results in the abstract refer to measurements of the initial 46 participants versus the 28 followed longitudinally.
A2. We have clarified which results indicated the initial 46 participants or 28 followed longitudinally (Page 1, Line 34-35) : New text
Q3. L34: “and the energy gap was significantly lower in the weight loss group than in the weight gain group (p < 0.05).” This may be clearer if the numeric values of the energy gap are reported, since it was positive for both groups but that was not stated.
A3. We have added the median values of the energy gap for both groups (Page 1, Line 36-37) : New text
Q4. L36: The abstract lacks a conclusion.
A4. We have corrected the conclusion for readers to understand our suggestion (Page 1, Line 37-39) : New text:
Q5. L53: “This may explain errors in the calculation of energy intake and malabsorption specific to older people, in addition to a lack of energy requirement.” I had difficulty interpreting this sentence. Do the authors mean that greater malabsorption in older people is not taken into account by the equations used to predict energy intake and, thus, the food provided is below their energy requirements?
A5 Thank you for your constructive suggestion. As you suggested, one of what we mean is that greater malabsorption is not taken into account by the equations for predict energy intake, thus, the food provided is below their energy requirements. Another thing we wanted to convey is that the facility staff serve the meal visually, thus, actual food provided may be less than energy of the recipe. Our manuscript seemed to have taken a leap in logic, and so we have corrected the paragraph 2 on Introduction (page 2, Line 56-67) : New text
Q6. L69: “First, they showed that the EI, determined based on dietary records, was overestimated by approximately 8%, but the weight change during the DLW period was not considered in the analysis, and it was unclear a discrepancy existed in the TEI. Second, they did not report the relationship between the energy gap and subsequent weight change; thus, it could not be determined whether the difference between the TEE and the calculated TEI was simply overestimation or actual overfeeding.” I am having trouble understanding the difference between the two points that the authors are trying to make here. Do the authors mean here that EI by food records was 8% higher than the TEE by DLW, but, since weight change was not reported, it is not clear whether some portion of this difference could be explained by weight gain (in the case of overfeeding) or whether the true difference between observed EI and measured EE may have actually been greater due to weight loss?
A6 Your comment is correct. We were confusing that two points. The first statement is related to whether the difference between the TEE and the calculated TEI was simply overestimation or actual overfeeding. Meanwhile, the second statement is related to your suggestion that it is not clear whether some portion of energy gap could be explained by weight change. We have corrected the manuscript (page 2, Line 78-83) : New text
Q7. L186: “First, the inter-rater reliability of the weight measurement between the researcher and facility staff was examined by intra-class correlation coefficient at baseline and 6 months and 12 month later.” Can the authors be more specific about how this was done? I assume the authors had both staff and researcher weigh all of the subjects included in analysis 2 at these time points, but it would be good to state that.
A7 Facility staff measure the weight for residents every month in Japanese long-term care facility. However, it is unclear how accurately these weights are measured because there is no manual for measurement of weight and their weight is assumed to be often measured with or without something in their pocket, wearing shoes and heavy jacket. Therefore, researchers also measured the weight of residents with light clothes, no shoes, and empty pockets, and we needed to compare both weights to examine the accuracy of the weight measured by facility staff. We have added above explanation (page 6, Line 202-208) : New text
Q8. L190: “It was hypothesized that the weight would change at a constant rate; therefore, the slope of the regression line of weight change over 12 months was used to assess the weight changes…” Can the authors be more specific about how often subjects were weighed? From Figure 2 it appears that they were weighed monthly, but some statement of this would help clarify.
A8 Weights were measured once a month by facility staff. We have added above explanation (page 6, Line 213) : New text
Q9. L192: “We also calculated the root mean square error of variation (RMSE) to support the above hypothesis.” By RMSE, are the authors referring to the RMSE between the regression line and the measured values? (as opposed to the baseline weight or mean weight or the 12-month period).
A9 The present study examined the relationship between energy gap and subsequent weight change, and so we needed to avoid including the resident who have lost weight during follow-up period because of acute illness. The RMSE was then calculated around each resident’s regression line for monthly weight changes and we confirmed their weight variability was small compared with a resident who was hospitalized during follow-up period. We have corrected the manuscript (page 6, Line 216-217) : New text
Q10. L194: “In the second analysis, the relationship between the energy gap and the slope of weight change was examined by Pearson's correlation coefficient.” Since all other analyses in the second group were non-parametric (presumably due to lower numbers) would it be more appropriate to use Spearman Rho instead of Pearson correlation (although the difference is likely small)?
A10 Before correlation analysis, we confirmed the normality of the energy gap and weight change using the Shapiro-Wilk Test and chose Pearson correlation. As you suggested, the result by Spearman Rho was similar to that by Pearson correlation (ρ=0.337, P=0.080 vs. r=0.357, P=0.063), and it seems easier to read manuscript by unifying the non-parametric method in analysis 2. Therefore, we have unified the non-parametric method in analysis 2 and corrected the manuscript (page 6, Line 218-219) : New text
Q11. L214: It should be noted that the plate waste rate data is in Table 1.
A11. We have added the plate waste data to into Table 2.
Q12. Table 2: the units for BMR should be kcal/d rather than kcal/kg/d.
A12. We have corrected the unit for BMR (Table 2).
Q13. Table 3: Can the authors elaborate on “Change in energy served during follow-up period”? Does this mean that the amount of energy served was changed for 2 people in each group? And in what way was it changed?
A13. As you suggested, the sentence means that the amount of energy served was changed for 2 people in each group. Among weight loss group, one resident increased energy served by 200 kcal/day because he walked around all day, and the other resident decreased energy by 250 kcal/day due to her wish. In contrast, in weight gain group, one resident increased energy by 150 kcal/day with the change of diet type, and another decreased energy by 200 kcal/day for weight control. These energy changes were not due to their health condition and not biased towards one group. Therefore, we considered that change in energy served during follow-up period was not relevant to weight change in this study. We have added the detail information of this result to manuscript (page 8, Line 271-275) : New text
Q14. L252: It may help the interpretation of the results to report the difference in OEI vs DLW-EI for the subset that were reported on in analysis 2 and by group in analysis 2 – i.e. was the overestimation of EI by observation similar at baseline for weight loss and weight gain groups?
A14. There was no significant difference for OEI-TEIDLW (kcal/d) between weight loss and weight gain groups (p=0.204). Therefore, the difference between energy gap (OEI-TEE) means that weight gain group ate more than weight loss group. We have added the results of the difference between OEI and TEIDLW into Table 2 and that explanation (page 8, Line 285-286) : New text
Q15. L269: “…and that the relationship between the energy gap and weight changes was not significant” The authors send a somewhat mixed message here, since in the abstract they note that “The energy gap tended to be correlated with the slope of weight change” and they point out that the energy gap is lower for weight loss group both in abstract and discussion.
A15. As you suggested, previous manuscript was inconsistent with abstract, result and discussion. We assumed that the energy gap would be significantly correlated with the slope of weight change but not significantly in actual. We consider that the above result was caused by small sample size because there was significant difference for the energy gap between weight loss group and weight gain group. Therefore, we unified the explanation of the result as follows: The energy gap tended to be correlated with the slope of weight change (page 10, Line 308-309) : New text
In addition, we have corrected the paragraph 1 on discussion to make the logically consistent (page 10, Line 312) : New text
Q16. L340: The authors could add the lack of measurement to determine malabsorption (stool and urine collection) and lack of follow-up measurements for TEE and OEI to limitations.
A16. In accordance with your comment, we have added two limitations (page 12, Line 385-389) : New text
Reviewer 2 Report
Interesting and important topic. The authors propose a very good rationale for this study. Very nice introduction. Please see below comments:
1. Relevant questionnaires used in methods. Would indicate to the reader more clearly which aspects of each questionnaire are relevant to the study, the data presented, and the outcomes/conclusion. For example, the MNA-SF was used. Did this indicate recent wt loss in the previous 3 months? or a decline in food intake previously?
2. Line 154, section 2.6 re calculation for TEI and comparison with OEI. Please clarify reference used. This doesn't seem to support the use of this equation.
3. A lot of data presented. Would include most relevant data to better 'tell the story' , and leave out data that is not clearly discussed/relevant to aims/conclusion.
4. Table 1: unclear why HB equation and PAL statement is included as Energy Setting? The authors state that indirect calorimetry was used to measure BMR. In general, Table 1 is confusing and recommend reworking. Please indicate the main point(s) of Table 1
5. Table 2: Is the TEE presented from DLW method or from measured BMR x PAL? Please clarify in table. Please also clarify in table how PAL was determined
6. Please clarify to the reader the main 'take away' from Table 3. Alot of information presented and unclear how these data contribute to the author's main objective and conclusion. Also please clarify if n=28? What is the significance of these data, particularly with such a low sample size? In addition, the author's note in the text that many participants could not stand, so is height estimated? please clarify this point. (in order to justify BMI, and the statement that BMI was not significantly different in these participants).
7. Table 4 seems to be the most important data. Please clarify TEE data presented (from DLW?) If so, please describe importance of BMR and PAL data presented. It is important for the authors to note that if n=28, only 8 participants exhibited wt loss
8. Please clarify n for Figure 3 (46?)
9. Please clarify the benefit/importance/relevance of weight measurements at 6 and 12 months without also looking at DLW TEI and OEI data at these time points also?
10. Recommend a stronger conclusion. What's the significance of line 342-342? What are the author's recommendation /future studies needed to address this OIE discrepancy and how does this apply (also in terms of feasibility) to the clinical setting in long term care facilities?
In general, very nice study with a lot of data. Would streamline data presented to better align with aims and conclusion. This study also seems an opportunity to discuss the need for more standardized and feasible methods for assessing energy needs in this population. The authors briefly mention the use of HB as a 'standard' when there are other better equations that can be used in this population. Thus- there seems to be the need to start using this equation vs. HB. In addition, how can these data apply to the 'real world'? DLW is of course the gold standard, is this feasible to use in measuring TEE in the clinical setting? What future studies can address ways to better estimate energy intake and even TEE in this population?
Author Response
Response to Reviewer 2 Comments
Reviewer 2
Interesting and important topic. The authors propose a very good rationale for this study. Very nice introduction. Please see below comments:
We would like to thank Reviewer 2 for the constructive suggestions to our manuscript intelligible. In previous manuscript, detailed information for dietary intake were shown by each facility in Table 1 because the sample size in this study was small and we were concerned about the influence of different calculation method for dietary intake among facilities. Actually, however, there seemed to be no bias that would affect the results or conclusion, and so we have moved Table 1 to supplements (Table S1). We also have revised the manuscript per the reviewer’s comments.
Our responses to the reviewer’s comments are as follows:
Q1. Relevant questionnaires used in methods. Would indicate to the reader more clearly which aspects of each questionnaire are relevant to the study, the data presented, and the outcomes/conclusion. For example, the MNA-SF was used. Did this indicate recent wt loss in the previous 3 months? or a decline in food intake previously?
A1. We thank the reviewer for this comment. One reason we showed the results of questionnaires was that it would be easier for the reader to imagine the characteristics of the present subjects. Barthel Index and MNA-SF is often used in the clinical setting while the Rapid Geriatric Assessment is easy to implement and can be used at long term care facilities. In addition, these questionnaires could be related to the energy gap. For example, Barthel Index, the FRAIL scale and the SARC-F scale would be related to physical activity and energy expenditure while MNA-SF and SNAQ would be related to the energy intake. We have described the relevance between questionnaires and main outcome (page 6, Line 193-195) : New text.
Q2. Line 154, section 2.6 re calculation for TEI and comparison with OEI. Please clarify reference used. This doesn't seem to support the use of this equation.
A2. Accordingly, we have clarified the reference used to calculate TEIDLW (page 5, Line 169) : New text.
Q3. A lot of data presented. Would include most relevant data to better 'tell the story', and leave out data that is not clearly discussed/relevant to aims/conclusion.
A3. We appreciate the reviewer's comment on this point. Table 1 was not directly relevant to results and moved to supplemental material (TableS1).
Q4. Table 1: unclear why HB equation and PAL statement is included as Energy Setting? The authors state that indirect calorimetry was used to measure BMR. In general, Table 1 is confusing and recommend reworking. Please indicate the main point(s) of Table 1
A4. As your comments, there was not enough explanation in original manuscript. We intended to show the calculation method for energy requirement used at each facility. Because Table 1 of original text has been moved to supplemental, we have corrected the Table S1.
Q5. Table 2: Is the TEE presented from DLW method or from measured BMR x PAL? Please clarify in table. Please also clarify in table how PAL was determined
A5. The TEE is presented from DLW method and PAL is calculated as the measured TEE divided by the measured BMR. We have added the explanation (page 7, Line 244, 247-248).
Q6. Please clarify to the reader the main 'take away' from Table 3. A lot of information presented and unclear how these data contribute to the author's main objective and conclusion. Also please clarify if n=28? What is the significance of these data, particularly with such a low sample size? In addition, the author's note in the text that many participants could not stand, so is height estimated? please clarify this point. (in order to justify BMI, and the statement that BMI was not significantly different in these participants).
A6. We thank the reviewer for this comment. Table 3 showed the variables that directly or indirectly related to energy expenditure and energy intake. We believe that these variables could be useful to determine whether the lack of energy served or healthy condition were responsible for weight loss. As you suggested, there were not enough explanation for the relevance between these results and objective. We have described that explanation (page 11, Line 360-363).
We obtained heights from medical records but did not know how height was estimated or measured because of the lack of its information. Meanwhile, we measured the lengths of lower leg for only 28 residents who could be tracked to the end, which was not described in the previous manuscript, and estimated height from these values by a predictive equation. Estimated height was significantly correlated with height from medical records (r=0.87, p<0.01), and there were no significant difference for BMI by estimated height between weight loss and weight gain groups (p=0.19). Therefore, height from medical records can be used. We have described the explanation (p 4, L 125-126) : New text
Q7. Table 4 seems to be the most important data. Please clarify TEE data presented (from DLW?) If so, please describe importance of BMR and PAL data presented. It is important for the authors to note that if n=28, only 8 participants exhibited wt loss
A7. We strongly appreciate the reviewer's comment on this point. The TEE presented measured value by the DLW method and that was clarified (Table 3) : New text. There was a significant difference for measured BMR between weight loss and weight gain groups but not significant for PAL. Moreover, Height and weight, which are related to BMR, were significantly higher in weight loss group than in weight gain group. From the above, the estimated energy requirement may be insufficient for residents with larger body size. We have added the sentence for importance of BMR and PAL, and also corrected the conclusion (p 8, L 287) : New text
Q8. Please clarify n for Figure 3 (46?)
A8. In accordance with your comment, we have added the number (page 8, Line 251) : New text
Q9. Please clarify the benefit/importance/relevance of weight measurements at 6 and 12 months without also looking at DLW TEI and OEI data at these time points also?
A9. The DLW method is quite expensive to implement and so we could not follow-up survey by the same measurement as baseline. Thus, body weights had been followed up to examine whether energy served was appropriate for elderly residents. We also obtained monthly weight data from medical records and used for analysis. However, it is unclear how accurately these weights are measured because it is assumed to be often measured with or without something in their pocket, wearing shoes and heavy jacket. Therefore, researchers also measured the weight of residents with light clothes, no shoes, and empty pockets and compared with the data from medical records. Considering that the weight change was relatively stable, we could obtain some implications described in this paper. We have added the explanation on the manuscript (page 3, Line 119-122) : New text
Q10. Recommend a stronger conclusion. What's the significance of line 342-342? What are the author's recommendation /future studies needed to address this OIE discrepancy and how does this apply (also in terms of feasibility) to the clinical setting in long term care facilities?
A10. We appreciate the reviewer's concerns on this point. As shown A7, the more energy served needs to be provided for residents who had larger body size. We have added the conclusion (page 12, Line 396-397) : New text
Q11. In general, very nice study with a lot of data. Would streamline data presented to better align with aims and conclusion. This study also seems an opportunity to discuss the need for more standardized and feasible methods for assessing energy needs in this population. The authors briefly mention the use of HB as a 'standard' when there are other better equations that can be used in this population. Thus- there seems to be the need to start using this equation vs. HB. In addition, how can these data apply to the 'real world'? DLW is of course the gold standard, is this feasible to use in measuring TEE in the clinical setting? What future studies can address ways to better estimate energy intake and even TEE in this population?
A11. We appreciate the reviewer's concerns on this point. At first, we were trying to clarify the energy requirement for elderly residents. However, the present result suggested the underestimation of OEI, which was unexpected result for us. We consider that it is better to discuss the energy need after clarifying the cause of underestimation because the facility staff will be confused. Meanwhile, we validated the simple physical activity record for estimating TEE compared with the DLW method in the previous paper (Nishida, JNSV, 2019). If use that simple record instead of the DLW method, we can increase sample size and examine the true energy requirement that takes into account the underestimation of OEI. We have added that sentence about future studies (page 12, Line 389-392) : New text